# Sebaceous Neoplasms

**DOI:** 10.3390/diagnostics13101676

**Published:** 2023-05-09

**Authors:** Ilias Papadimitriou, Efstratios Vakirlis, Elena Sotiriou, Katerina Bakirtzi, Aimilios Lallas, Demetrios Ioannides

**Affiliations:** First Department of Dermatology and Venereology, School of Health Science, Aristotle University of Thessaloniki, 54643 Thessaloniki, Greece

**Keywords:** sebaceous neoplasms, sebaceous carcinoma, sebaceoma, sebaceous adenoma, sebaceous hyperplasia, Muir–Torre Syndrome

## Abstract

Sebaceous neoplasms describe a group of tumors with sebaceous differentiation commonly seen in lesions located primarily in the face and neck. The majority of these lesions are benign, while malignant neoplasms with sebaceous differentiation are uncommon. Sebaceous tumors present a strong association with the Muir–Torre Syndrome. Patients suspected with this syndrome should undergo neoplasm excision, followed by histopathologic and additional immunohistochemistry and genetics examinations. Clinical and dermoscopic features of the sebaceous neoplasms, as well as management procedures collected from the literature analysis regarding sebaceous carcinoma, sebaceoma/sebaceous adenoma, and sebaceous hyperplasia are described in the current review. A special note is made for describing the Muir–Torre Syndrome in patients presenting multiple sebaceous tumors.

## 1. Introduction

Sebaceous differentiation of neoplasms is represented by sebaceous carcinoma, sebaceoma, adenoma, and sebaceous hyperplasia. The benign sebaceous gland hyperplasia is very common among the general population, while malignant neoplasms, sebaceous carcinoma, sebaceoma, and adenoma are very rare. Sebaceous gland tumors can appear at any age, but they affect mainly elderly people. The group of sebaceous tumors have been strongly associated with Muir–Torre Syndrome (MTS)—when multiple lesions are present—while sebaceous hyperplasia has been correlated with chronic sun exposure. In this review, an effort was made to describe the clinical and dermoscopic features of sebaceous neoplasms, along with the management of such lesions, regarding tumor excision and the presence of MTS.

Clinically, sebaceous tumors are often mistaken for more common malignant skin neoplasms, such as basal cell carcinoma and squamous cell carcinoma. In these cases, the diagnosis can be assisted by the use of dermoscopy and confirmed with the help of histopathology. Sebaceous tumors are mainly located on the head and neck area, but they may be traced to any region of the body. The clinical presentation can vary from small, skin-colored, umbilicated papules to ulcerated lesions, subcutaneous nodules, or exophytic tumors. The lesions are solitary except in patients with MTS. In this case, numerous sebaceous tumors may be located outside the head and neck area. Moreover, patients affected from the syndrome can present multiple keratoacanthomas, or manifest gland neoplasms at a young age.

## 2. Materials and Methods

For this review, we searched the following databases: PubMed, MEDLINE, Scopus, OVID, Embase, Science Direct, Cochrane library, Research Gate, and Google Scholar. The search terms were “sebaceous neoplasms” OR “sebaceous carcinoma” OR “sebaceoma” OR “adenoma” OR “sebaceous hyperplasia” OR “Muir–Torre Syndrome”. Studies were included from inception to February 2023. Data extraction was performed individually for each neoplasm. The pertinent information obtained from each study was: the first author; year of publication; type of study; type of sebaceous neoplasm; epidemiologic data; clinical features; dermoscopic features; and management. The search included exclusively English-language academic papers. The reference list of the shortlisted articles was also examined for supplementary studies. The study has been designed, and the results have been described as per the Preferred Reporting Items for Systematic Reviews and Meta-Analyses (PRISMA) statement. Our literature search yielded 494 studies. After excluding case reports, systematic reviews, duplicate studies, and studies that did not present any relevant data, we ended up with 52 studies.

## 3. Epidemiology of Sebaceous Neoplasms

Sebaceous carcinoma is an uncommon cutaneous neoplasm. The prevalence reported in two studies, based on the US and Texas registries of skin malignancies, varies between 0.05% and 0.7% among other skin tumors [1,2]. Although uncommon, it represents 1–3.2% of all periorbital malignancies [3,4]. Risk factors mentioned in the literature are elderly age, with a mean age of 70 years, and Asian race [2,3,5,6]. Of note, sebaceous carcinoma is less frequent in Africans with an incidence of 0.48 per million [3].

In a 9-year-study about sebaceous neoplasms, from a total of 8819 biopsies, 207 sebaceous neoplasms were isolated. Among them, 1.6% were identified as sebaceomas and 1.1% as adenomas [7]. Sebaceomas and adenomas, like sebaceous carcinomas, demonstrate a preference for the elderly, but they may occur at a younger age too. Sebaceous hyperplasia was the most common neoplasm identified, with 35.1%, where sebaceous carcinomas were found in 14.3% of the total amount of sebaceous tumors [7].

## 4. Pathogenesis and Genetic Aspects of Sebaceous Tumors

The pathogenesis of sebaceous tumors is poorly understood, such as the genetic mutations correlated with these tumors. However, it is well established that chronic sun exposure, irradiation, and induced or acquired immunosuppression are strongly associated with the appearance of sebaceous neoplasms. Apart from these conditions, familiarity with retinoblastoma also seems to be linked to malignant sebaceous differentiation epitheliomas. Well established is the association between sebaceous carcinoma and MTS, a variant of Lynch syndrome with a presence of skin neoplasms [2,8]. The variations of the normal signaling pathway Wnt/beta-catenin are strongly correlated with the abnormal and malignant cell proliferation of the sebaceous gland. The transcription enhancer-binding factor (LEF-1) mutations seem to play a critical role in the development of skin tumors. Sebaceous carcinomas may present a complete silence of the LEF-1 gene, while both sebaceomas and sebaceous adenomas can also present mutations of the gene [8]. Moreover, the dysregulation of the transforming growth factor-beta, PTEN, and nuclear factor-kappaB, as well as the inactivation of p53, was found in some sebaceous carcinomas [9]. The increased expression level of tyrosine kinase and especially HER2 has been associated with sebaceous tumors and investigated as a possible targeted treatment [10].

## 5. Sebaceous Carcinoma

Sebaceous carcinoma is an adenocarcinoma demonstrating variable levels of sebaceous differentiation. These epitheliomas demonstrate a preference for appearing in the eyelid and are typically classified as periocular and extraocular. This anatomic classification is accompanied with various genetic mutations and tumor aggressiveness; moreover, there is a tendency for diffuse intraepithelial growth (pagetoid spread) [7,9,11,12]. In fact, periocular tumor presence may be accompanied with widespread metastasis, while metastasis is infrequent for the extraocular type [13,14,15]. Extraocular appearance is uncommon and includes tumor positioning on the head and neck, while it can be rarely found on the foot, penis, and vulva. Sebaceous carcinoma has a 30 to 40% risk for local tumor recurrence, up to 20% for distant metastases [16]. The 5-year overall survival rate is 78% for the localized disease and 50% for the metastatic [17]. However, with increased disease awareness and the application of advanced excision procedures, the survival rate has been significantly improved over the last years [18]. Selective tumor positioning on the head, and especially in the periocular area, is the cause for increased treatment-related morbidity, as extensive facial reconstruction techniques are performed [19]. The presence of sebaceous carcinoma can be sporadic in the context of MTS.

Periocular sebaceous carcinoma is very common in Asia, representing 40–60% of eyelid tumors. However, in the western world, it is extremely uncommon, making up only 1% of periorbital tumors [8,20,21]. The incidence of sebaceous carcinoma over the last decade presents an annually increasing rate of 3.31%, followed by increased mortality and extraocular location occurrence rates [19]. The peak incidence for all race and sex groups is reported between the ages of 60 and 79 years [19]. Etiology of the disease includes irradiation, immunosuppression, exposure to nitrosamines, and the presence of the MTS.

The periocular tumor typically arises from the tarsal glands, usually in elderly patients, and it is commonly located in the upper eyelid. The neoplasm may affect one or both eyelids and conjunctiva. Clinically, it appears as a yellow to pink, painless, rapidly enlarging, firm papule, nodule, or cystic lesion (Table 1). Dermoscopy reveals the presence of polymorphous atypical vessels and an underlying yellow background. The presence of ulceration suggests the malignancy of the neoplasm [16]. In the early phase, it can be easily mistaken for chalazion, blepharocunjuctivitis, or keratoconjunctivitis. In more advanced cases, it may lead to ulceration and distorted vision. If left untreated, metastasis to local lymph nodes and parotid gland is not infrequent.

The extraocular tumor is infrequent, representing 25% of all cases of sebaceous carcinoma [22]. Clinically, it appears as a yellow to pink or red firm nodule of different sizes. Contrary to the periocular type, the clinical feature of the extra ocular neoplasm is non-specific. Applying dermoscopy, we can distinguish a number of atypical vessels and yellowish lobules.

Establishing the diagnosis of suspicious neoplasms for sebaceous carcinoma differs upon localization. In extraocular neoplasms with a clinical and dermoscopic suspicion of sebaceous carcinoma, a deep biopsy including dermis should be performed. In periocular lesions with a clinical and dermoscopic image compatible for sebaceous carcinoma, a superficial biopsy—inclusion of the dermis is not required—should be performed. Sebaceous carcinoma should be included in the differential diagnosis in cases with persistent unilateral blepharitis or chalazion, especially in patients over the age of 60 [22].

Achieving the diagnosis of sebaceous carcinoma can be difficult even for pathologists. The routine hematoxylin–eosin-embedded sections demonstrate a basoloid neoplasm in lobules or sheets of cells separated by a fibrovascular stroma with infiltrating edges, while cytologically hyperchromatic nuclei and prominent nucleoli are observed [4]. The degree of cellular differentiation can vary and is classified as poorly, moderately, and well differentiated. Pagetoid tumor growth is commonly observed in periocular lesions, while in extraocular locations it is an unusual finding. Squamous metaplasia or focal apocrine differentiation can also be noted in some cases. According to WHO, three levels of grading are proposed: well-demarcated tumors with roughly equally sized cellular lobules are classified as grade I; those presenting a mix of well-defined nests and infiltrative features and/or confluent nests as grade II; and finally, grade III are those with a highly invasive growth and/or medullary sheet-like pattern [23,24]. The tumor shares common findings with the basaloid squamous carcinoma, and both types are capable of sebaceous differentiation. Moreover, sebaceous carcinoma may simulate basal cell carcinoma with less differentiated neoplastic cells containing basophilic cytoplasm and mitotic figures at the periphery of an infiltrating lobule. Immunohistochemistry may be a reliable help for distinguishing the sebaceous tumor from the basaloid squamous carcinoma, basal cell carcinoma, and even from some types of melanomas. The main markers used are the epithelial membrane antigen (EMA) and the monoclonal antibody directed toward the epithelial cell adhesion molecule Ber-Ep4 [25]. The EMA is predominantly positive in sebaceous tumors and squamous cell carcinomas, while it is negative for the basal cell carcinomas. On the other hand, sebaceous carcinomas and squamous cell carcinomas are negative for Ber-Ep4 in 74–94% and 100% of cases, respectively, and BCC shows positive immunostaining in 70–100% of cases [26]. The marker Terminal deoxynucleotidyl Transferase (TdT), a DNA polymerase expressed in immature, normal and neoplastic, lymphoid, or haematopoietic cells, and in neuroendocrine carcinomas, such as Merkel cell carcinoma and small-cell carcinoma, has been also found to mark epithelial cells with sebaceous differentiation, both malignant and benign. Negative staining for immunoperoxidase techniques such as S100, HMB-45, melan-A (MART-1), and NKI/C3 can be of great help in distinguishing sebaceous carcinomas from melanomas, with the exception of melanomas with little or no pigmentation or spindle cell growth pattern. PRAME immunostaining, mainly used in immunohistopathology for the differential diagnosis between dysplastic melanocytic nevi and malignant melanomas, has been used lately as an immunomarker for the sub-classification of sebaceous carcinomas in grade I, II, and III [23,27]. It has been reported that PRAME can be useful in the subclassification of sebaceous carcinomas, highlighting the foci of mature sebaceous differentiation most present in grades I and II, and almost completely absent in grade III [23]. Other neoplasms that can mimic the sebaceous carcinoma are the sebaceous benign neoplasms, the endocrine mucin-producing sweatgland carcinoma, the spiradenocarcinoma, the clear cell hidradenocarcinoma, and finally the great mimicker poroma.

The management of extraocular sebaceous carcinoma is based on complete tumor excision with histological margin control. Both Mohs micrographic excision and complete circumferential peripheral and deep margin assessment (CCPDMA) represent the gold standard techniques, followed by wide local excision, in order to prevent recurrence [22]. The latter should be performed with a margin up to 0.5 cm and down to the fascial plane [28]. Destructive techniques are not recommended due to the tumor malignancy and tendency to relapse or metastasize. Radiotherapy as monotherapy is also not recommended and should be limited to inoperable cases [22]. In such cases, given the weak evidence of radiation dosing, it is preferable to consult with a radiation oncologist with experience in sebaceous carcinoma and then proceed to a close follow-up of the patient. Adjuvant radiotherapy might be considered in cases with perineural invasion when excisional margins are positive [22]. Radiotherapy of lymph nodes can be considered, with unclear results on the overall increase in survival. Regional–sentinel–lymph node biopsy should be recommended and limited to the cases with clinical lymph node enlargement [22].

Periocular neoplasm management, due to its anatomical position, is more demanding. Histological margin excision techniques such as Mohs and CCPDMA are preferable over wide local excision. When margin control excision is not possible, staged excision with delayed reconstruction can be considered, while in cases of positive margins, the use of cryotherapy and mitomycin-C may be considered, but limited data exist [20,29,30,31,32]. Data regarding radiotherapy as monotherapy present ambiguous results. Sessions of radiotherapy with doses over 55 GY were associated with better disease control, but in a limited number of cases [33]. Adjuvant radiotherapy has been used when the lacrimal system or the orbit has been involved instead of exenteration [11,34,35]. A sentinel lymph node biopsy with or without imaging of the regional nodes can be considered for periocular sebaceous carcinoma at stage T2c or higher [22].

The tumor may present regional and distant lymph node metastasis, as well as pulmonary, hepatic, bone, and cerebral ones. Systemic therapy for the management of metastasis includes conventional chemotherapy, immunotherapy, or targeted therapies such as antiandrogen, retinoid receptor ligands, and EGFR [22].

A close clinical examination with a follow-up every six months should be performed for the first three years following the treatment. The clinical examination should be accompanied by an ultrasonographic evaluation of the regional lymph nodes, or with CT and PET-CT in advanced disease cases.

The early diagnosis of sebaceous carcinomas is considered the key stone for disease-free patients, along with surgical excision with free, histopathological controlled, margins, by performing Mohs or CCPDMA. When disease free margins are not possible, the usage of adjuvant radiotherapy or topical destructive techniques could be of use in order to delay or avoid exenteration. Moreover, screening for the possible arise of Muir–Torre Syndrome should be considered.

## 6. Sebaceoma and Sebaceous Adenoma

Sebaceoma and sebaceous adenoma are both benign skin neoplasms with sebaceous differentiation. Clinically, they appear as solitary or multiple nodules of skin or a yellowish or reddish color, while their surface can present a multi-lobule appearance (Figure 1). Ulceration is non-typical for these adenomas and, when presented, malignancy should be suspected [36]. Most lesions are of a small size with a diameter of less than 1 cm, although lesions up to 5 cm have been documented. Commonly affected areas are those rich in sebaceous glands, such as the head and neck. Most lesions are asymptomatic and slow growing.

Sebaceoma and sebaceous adenoma represent the same clinical feature, but with different histopathologic findings. In fact, the term sebaceoma is used when the presence of basaloid cells is more prominent, with more than 50% of the tumor cells being basaloid. On the other hand, when the number of basaloid cells is less than 50% the tumor is called sebaceous adenoma. Sebaceoma histology is presented as a dermal nodule with variable epidermal involvement consisting of basaloid cells and mature sebocytes. Histopathologic variants of sebaceoma have been described with carcinoid-like, reticulated, cribriform, rippled, and Verocay body-like features, but the clinical appearance remains unaltered. Sebaceous adenoma presents an image of well-circumscribed lobules with a nodular growth consisting of an admixture of basaloid cells and mature sebocytes. Some lobules may communicate directly with the surface epithelium. Basaloid cells are usually located at the periphery of lobules, while sebaceous cells with intracytoplasmic lipid vacuoles are usually located at the center of lobules.

The dermoscopic features of sebaceomas and sebaceous adenomas are overlapping, rendering the diagnosis between them impossible, but this has meaningless clinical significance. Both tumors can present two distinctive patterns, with or without a central depression as a crater. Lesions with a central crater present elongated crown vessels around a structureless yellow or white ovoid center, which at times can be covered with blood crusts (Figure 2) [37,38,39,40,41]. Tumors without a central crater present arborizing vessels over a white or yellowish background and occasionally yellow comedo-like globules [38]. The branching vessels of these adenomas are unfocused, distinguishing them from the ones found on basal cell carcinomas.

The differential diagnosis of sebaceomas and sebaceous adenomas include the basal cell carcinoma; the trichoblastoma; adnexal tumors with sebaceous differentiation; and the panfolliculoma. In rare cases, sebaceous adenoma may be difficult to distinguish from well-differentiated sebaceous carcinoma. In order to differentiate from the basal cell carcinoma and the trichoblastoma, immunohistochemistry may be of a great help. The marker Ber-Ep4 is positive for the aforementioned neoplasms, while sebaceomas and sebaceous adenomas are almost always negative.

A complete surgical excision of these benign lesions is recommended, over a punch biopsy or other destructive methods, in order to achieve diagnosis and avoid the misdiagnosis of the well-differentiated sebaceous carcinoma and the rare basal cell carcinoma with sebaceous differentiation. The pathology is also essential in order to evaluate the patient for the existence of possible Muir–Torre Syndrome and to set an annual surveillance for visceral and cutaneous malignancy strategy [42].

## 7. Sebaceous Hyperplasia

The most common proliferative abnormality, arising in one of every four adults is represented by sebaceous hyperplasia [43]. The lesions arise from anatomical sites rich in sebaceous glands—such as the face—and are often multiple, even though solitary lesions can be also found. Clinically, they appear as firm, skin-colored or whitish–yellow, asymptomatic, and slowly growing umbilicated papules, with a diameter that is usually less than 1 cm [16,44] (Figure 3). Sebaceous hyperplasia can also be observed in full-term neonates but, in this case, it is transient as a result of the neonate exposure to maternal hormones [36,45,46]. High doses of cyclosporin and antiretroviral drugs can induce hyperplasia of the sebaceous glands [36].

The dermoscopic features of sebaceous hyperplasia are the white–yellowish globules—in the form of cumulous clouds—surrounded by small crown vessels. The follicular ostia is commonly visible as a central umbellic (Figure 4). The presence of both features is valuable for the establishment of the diagnosis.

Pathology reveals an increased size gland that fully retains the normal cellular architecture. The lobules are attached to the hair follicle, while the presence of at least four lobules per hair follicle is suggested as a diagnostic criterion [36]. Another commonly used criterion, this time for distinguishing sebaceous hyperplasia from sebaceous adenoma, is the number of germinative cell layers in the periphery of the lobules; when more than two germinative cell layers are present, a sebaceous adenoma should be suspected [47,48].

When the differential diagnosis poses dilemmas, a biopsy should be performed to exclude the rest of the sebaceous gland neoplasms and adnexal tumors. In order to establish the diagnosis, a total excision should be performed since, in some cases, the distinction between sebaceous adenoma and sebaceous hyperplasia is not clear-cut, especially if the histopathologist has to deal with shave biopsies or incomplete excisions. Establishing the diagnosis and excluding other sebaceous neoplasms is very important, as sebaceous hyperplasia has no correlation with MTS. The rest of the differential diagnoses include the sebaceous nevus; the basal cell carcinoma; the molluscum contagiosum; and finally, the xanthoma.

Even though sebaceous hyperplasia is a benign lesion, it can cause a significant psychological burden since the lesions are almost always multiple and positioned on the face, posing a significant cosmetic concern. The treatment approach can also be tricky, since the sebaceous gland lies deep into the dermis, and the usage of destructive techniques can cause scarring or relapse of the hypertrophic gland. Traditional treatments for sebaceous hyperplasia include cryosurgery, electrodessication, curettage, shave excision, and topical trichloroacetic acid, with the risk of post-treatment hypo or hyper pigmentation [49]. Topical photodynamic therapy seems to have optimal results, over other light-source therapies. The systemic administration of retinoids is effective, but is associated with a relapse of the hyperplasia upon treatment discontinuation, and the adverse effects of the systemic administration of isotretinoin [45]. Both can be avoided with intermittent or low doses of isotretinoin after the disease remission.

## 8. Muir–Torre Syndrome

Muir–Torre Syndrome was first described in 1967 by Dr. E.G. Muir, and one year later by Dr. Douglas Torre. MTS is a rare phenotypic type of hereditary non-polyposis colorectal cancer (HNPCC) which is also known as Lynch syndrome [50]. MTS is a rare autosomal dominant disorder, defined by the presence of multiple skin neoplasms (mainly with sebaceous differentiation) and the presence of visceral malignancies, mainly colorectal, endometrial, ovarian, and renal/pelvis/ureter carcinomas. The cutaneous neoplasms that have been related to MTS are the following: sebaceous adenomas; sebaceous epitheliomas; sebaceous carcinomas; cystic sebaceous tumors; basal cell carcinomas with sebaceous differentiation; and keratoacanthomas. It is caused by microsatellite instability due to mutations in DNA mismatch repair proteins. More recently, a subtype of MTS was reported that does not demonstrate microsatellite instability and is due to defects in the base excision repair gene known as MYH. This variant is estimated to be responsible for 35% of all cases with MTS, and is inherited with an autosomal recessive pattern [51]. Sporadic cases of MTS caused by immunosuppression with tacrolimus and cyclosporine have also been reported in the literature [52,53]. Suspicion of MTS can rise during a dermatologic visit and, if confirmed from the pathology, immunohistochemistry, and genetic testing, it requires close cancer surveillance for the patient and for the whole family, due to the increased tendency of those patients to develop visceral malignancies and sebaceous carcinomas.

MTS is seen in 9.2% of individuals affected with HNPCC [54]. Most reported cases are related to Caucasians, probably due to a lack of epidemiologic data from Asia and Africa. The onset of malignancy ranges from 23 to 89 years with a median age of approximately 53 years [55]. The majority of MTS patients present germline pathogenic variants in DNA mismatch repair (MMR) genes: the mutL homolog1 (MLH1), mutS homolog2 (MSH2), mutS homolog6 (MSH6), and postmeiotic segregation increased2 (PMS2). In addition, germline deletions within the epithelial cell adhesion molecule gene have also been implicated in Lynch syndrome, as these deletions can disrupt the MMR pathway through the inactivation of the adjacent MSH2 gene [56]. The four MMR genes are responsible for encoding the four MMR proteins, respectively. These proteins form pairs with each other in order to detect and repair errors in DNA replication; MLH1 dimerizes with PMS2, while MSH2 dimerizes with MSH6 [57]. Germline alteration in one of the MMR genes followed by a second somatic hit to the remaining wild type allele leads to genomic microsatellite instability (MSI) which, in turn, predisposes the development of malignancies [58]. It has been demonstrated that the sebaceous neoplasms associated with MTS show a loss of MSH2 expression in over 90% of cases, followed by a loss of MLH1 [59,60,61]. Only some limited cases with a loss of MSH6 or PMS2 have been documented in the literature [59]. The pathogenic gene variants can be indirectly evaluated in the neoplasms by using immunohistochemistry. With this method, it is possible to evaluate the protein expression of the genes. The evaluation of the immunohistochemistry results is difficult, however, as the MMR proteins form dimers, and may have multiple binding partners.

Currently, the diagnosis of MTS is based on genetics, as the clinical diagnostic criteria are ambiguous. The immunohistochemistry performed on sebaceous excised neoplasms can lead to diagnosis with great accuracy, but it can also be inaccurate in determining the MTS due to the possible presence of somatic, rather than germline, mutations. In such cases, a detailed personal medical and family history could be of great help. The Mayo MTS risk scoring algorithm may be used to identify patients who should undergo further germline MMR genetic testing (Table 2) [62]. When immunohistochemistry results are positive, the diagnosis should be confirmed with genetics, prior to proceeding with the extended visceral malignancy screening.

Once the diagnosis has been established, MTS patients should undergo annual surveillance for visceral and cutaneous malignancy. Gastrointestinal endoscopy should be performed annually, with colonoscopy examination beginning from the 18th year of age, and upper endoscopy from the 25th [50]. Men should undergo annual prostate and testicular examinations, while women should have annual breast and pelvic examinations along with a transvaginal ultrasound and endometrial sampling. Other tests include chest X-ray, carcinoembryonic antigen, blood count, and liver function control.

The management of cutaneous lesions in the context of MTS does not differ from that of solitary neoplasms and was described earlier in the current review.

## 9. Conclusions

Neoplasms with sebaceous differentiation present an everyday challenge for the clinician. In cases of benign neoplasms that clinically present relevant difficulty regarding the differential diagnosis, the use of dermoscopy could be of a great help. In order to establish the diagnosis, histopathology should be performed as neoplasm differentiation variations could lead to misdiagnosis and mistreatment. Moreover, since the differential diagnosis should include the sebaceous carcinoma, the excision management could differ significantly. The therapeutical approach in sebaceous carcinomas should include histological margin control excision techniques, as the rate of relapse or metastasis is significantly high. When this is not applicable, wide margin excision limits should be granted. Association with Muir–Torre Syndrome should also be considered in all cases with multiple tumors with sebaceous differentiation. Immunohistochemistry may provide useful information in cases where histologic distinction is difficult, mainly between sebaceous adenoma and sebaceous hyperplasia. This dilemma should be confronted since sebaceous hyperplasia is not associated with MTS, whereas sebaceous adenoma is the most common sebaceous tumor in the syndrome. The use of genetics is the gold standard in order to detect the germline mutation and achieve the diagnosis of MTS. In MTS patients, a multidisciplinary approach is essential, while close follow-up should be performed.

Understanding the nature and the behavior of sebaceous tumors, defying the overlap between the various neoplasms with sebaceous differentiation, describing and improving the clinical and dermoscopic criteria for the diagnosis, and investigating the possible presence of immunohistochemistry germline mutation markers should be subject to further studies. Since clinical or dermoscopic definitive diagnosis is not achievable, all sebaceous tumors should be treated with caution, and excised totally.

## Figures and Tables

**Figure 1 diagnostics-13-01676-f001:**
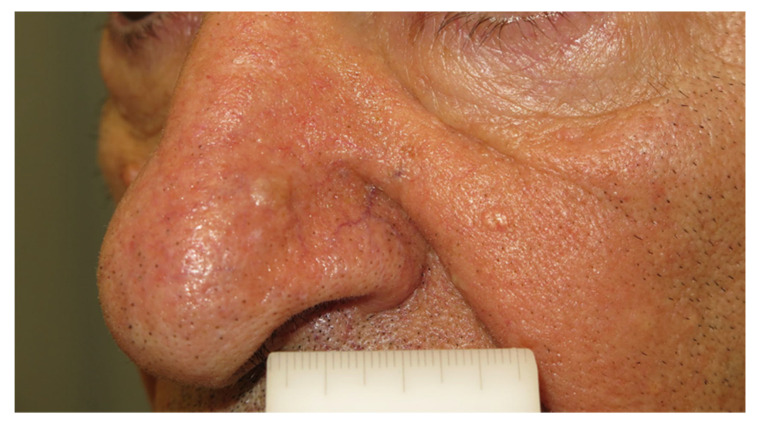
Multiple skin-colored papules on the nose and the left cheek, without signs of ulcerations, sebaceous adenoma.

**Figure 2 diagnostics-13-01676-f002:**
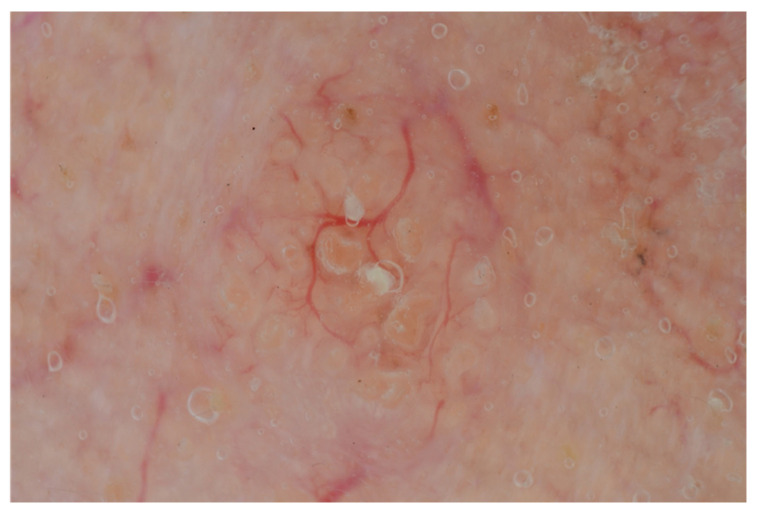
Dermoscopic image of sebaceous adenoma demonstrating arborizing and crown vessels, yellowish comedo-like structures, without ulceration.

**Figure 3 diagnostics-13-01676-f003:**
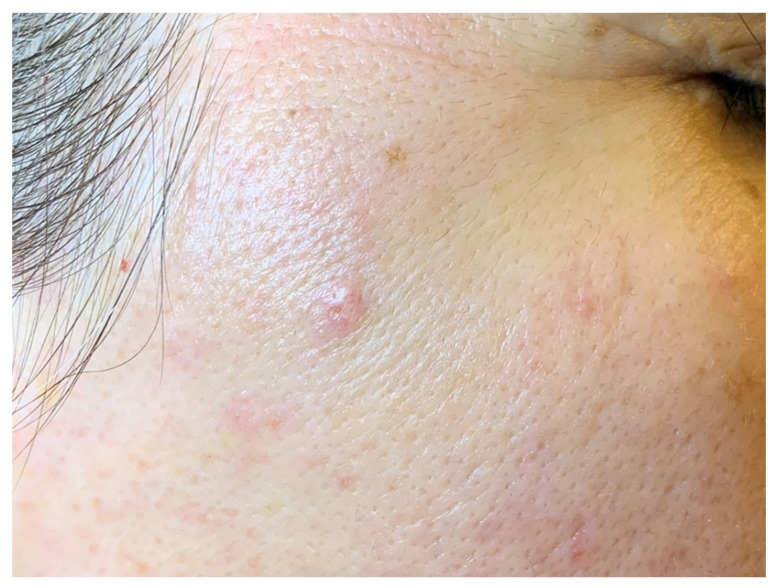
Solitary, small, skin-colored, umbilicated papule arising in the region of zygomatic bone, sebaceous hyperplasia.

**Figure 4 diagnostics-13-01676-f004:**
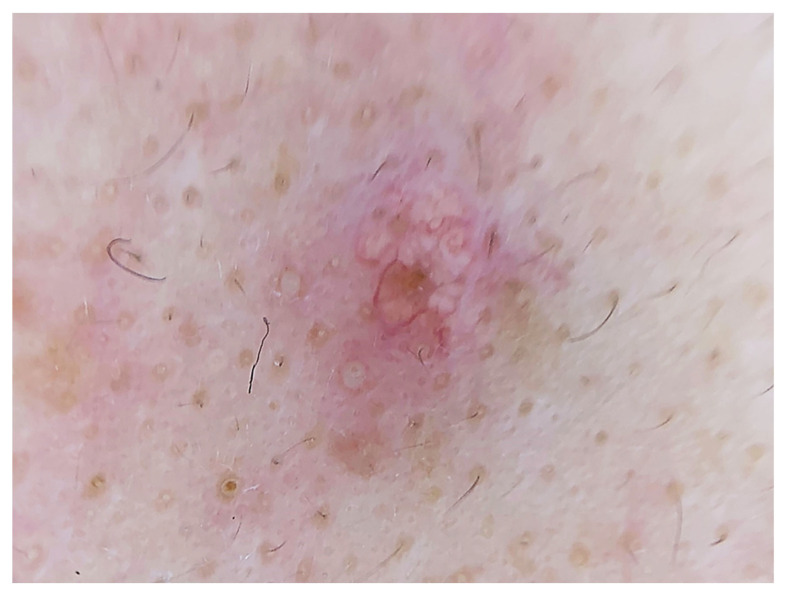
Dermoscopic image of sebaceous hyperplasia demonstrating yellowish cloudy formations with small vessels surrounding the sebaceous ostia.

**Table 1 diagnostics-13-01676-t001:** Clinical and dermoscopic findings of sebaceous neoplasms.

Neoplasm	Clinical	Dermoscopic
Sebaceous carcinoma	Yellow to pink, painless, rapidly enlarging, firm papule, nodule or cystic lesion, ulceration	Polymorphous atypical vessels mainly peripheral, underlying yellow background, ulceration
Sebaceoma/Sebaceous adenoma	Skin or yellowish or reddish color nodules, multi-lobule surface	Two distinctive patterns: (a) Umbilicate form: elongated crown vessels around a structureless yellow or white ovoid center, central blood crusts (b) Non umbilicate: arborizing vessels (non in focus), white or yellowish background, yellow comedo-like globules
Sebaceous hyperplasia	Firm, skin colored or whitish–yellow umbilicated papules, less than 1 cm diameter	White-yellowish globules—in form of cumulous clouds, peripheral small crown vessels, visible follicular ostia as a central umbellic

**Table 2 diagnostics-13-01676-t002:** The “Mayo MTS risk score”.

Variable	Score
Age at sebaceous neoplasm 1 diagnosis (years)	
60 or older	0
Less than 60	1
Total number of sebaceous neoplasms	
1	0
2 or more	2
Personal medical history of lynch-related cancer 2	
No	0
Yes	1
Family medical history of lynch-related caner 2	
No	0
Yes	1

Total scores for each variable is summed to create the “Mayo MTS risk score,” ranging from 0 to 5. A score of 2 or more has a sensitivity of 100% and specificity of 81% for predicting a germline mutation in a Lynch syndrome MMR gene. ^1^ Mayo MTS risk score applicable to patients with sebaceous adenomas, sebaceous epitheliomas, sebaceomas, and sebaceous carcinomas. ^2^ Lynch-related cancers include CRC, endometrial, ovarian, small bowel, urinary tract, and biliary tract cancers.

## Data Availability

The data that support the findings of the study are available from the corresponding author, Vakirlis Efstratios, upon request.

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
