# Peer review of "Sebaceous Neoplasms"

_diagnostics, 2023, doi:10.3390/diagnostics13101676_

Round 1

Reviewer 1 Report

Dear Authors;

When the literature was reviewed, there was no good and useful comprehensive review article on sebaceous tumors recently (in the last 3-4 years). The present article is a well-written article where information about sebaceous tumors can be accessed collectively. Current literature is included in the article. It can be published in your journal, can be cited and can contribute to the literature. But, If, a pathologist and histopathological and immunohistochemical pictures also may be included, It would be great. but it is not essential. sincerely

Author Response

Dear Reviewer,

Thank you for your comments!

In order to enhance the value of the Review, we proceeded by adding supporting histopathologic details as well as some immunohistochemical information (especially for the sebaceous carcinoma) accompanied by corresponded references.

Unfortunately, it is impossible to add histopathologic or immunohistochemical figures within the period of 10 days given to us for submitting the revised paper, as this would require the cooperation of the pathology department, that is currently under heavy workload. We kindly aspect for your comments.

We would like to thank you again for taking the time to review our manuscript. We look forward to hearing from you regarding our submission. We are at your disposal to promptly respond to any further questions and comments you may have.

Sincerely,

Ilias Papadimitriou

Reviewer 2 Report

I want to congratulate the authors on the submitted manuscript. The review done on this entity is very complete and useful in a multidisciplinary way.

However some revisions can make the manuscript more complete and usable by a greater number of readers:

- From a clinical point of view, sebaceous carcinoma does not present specific signs and symptoms, while histologically it presents more peculiar characteristics. The inclusion of histological figures with an accurate histological description of sebaceous carcinoma would make the work more valuable.

- Moreover, considering the diversity of histological presentations, this entity also enters in differential diagnosis with some forms of unusual melanomas (balloon cell, verrucous, signet ring cell, small cell, etc.). In these cases HMB45, MELAN-A, SOX10 or new markers such as PRAME may be useful for a differential diagnosis (DOI: 10.3390/diagnostics12092197). On the other hand, PRAME can be useful in the sub-classification of sebaceous carcinoma in grades I–II–III according to the directives of the latest WHO 2018 (DOI: 10.3390/jcm11236936).

Author Response

Dear Reviewer,

Thank for your fruitful comments, which were truly to the point.

In accordance with your comments, we proceeded by adding details about the pathology and the immunohistochemical staining. Unfortunately, it is impossible to supply the corresponding figures in the period of 10 days given to us in order to resubmit the review, because this would require the involvement of the pathology department of our hospital, which is currently under surprisingly heavy workload (probably a post-covid phenomenon!).

Regarding the second point, we would like to thank you for pointing out this information. We proceeded by adding the appropriate details and references kindly provided by you, as they really unveil some really useful clinical information for the differential diagnosis of sebaceous carcinoma.

We would like to thank you again for taking the time to review our manuscript. We look forward to hearing from you regarding our submission. We are at your disposal to promptly respond to any further questions and comments you may have.

Sincerely,

Ilias Papadimitriou

Round 2

Reviewer 2 Report

I congratulate the authors for the changes made to the manuscript.

I'm sorry they couldn't include a histological figure but I understand their motivations.

However this latest version satisfies the high quality criteria imposed by the magazine and I recommend its publication.